# Monitoring of the Behaviour and State of Nanoscale Particles in aGas Cleaning System of an Ore-Thermal Furnace

**Vladimir Bazhin and Olga Masko** *

The Automation of Technological Processes and Production Department, Saint Petersburg Mining University; 2, 21st Line, 199106 St Petersburg, Russia; bazhin-alfoil@mail.ru

* Correspondence: olgamasko.17@gmail.com or s205017@stud.spmi.ru

**Abstract:** The aim of this paper is to define and select stable zones in the off-gas duct of an ore-thermal furnace using a mathematical model. This is needed to increase the effectiveness of exhaust gas composition control in metallurgical silicon production. **Methods.** The goals of this study were achieved by means of computational fluid dynamics. A model with a water-cooled furnace roof as well as a model comprising steel gas passes with a sliding shutter was developed using ANSYS Fluent software. Both models were symmetrical to ensure a uniform gas-dust distribution, which allowed us to test the adequacy of the obtained models. The models were based on the Navier–Stokes equations system as well as on a discrete phase model (DPM) that was developed using the Euler–Lagrange method. **Results.** As a result of the modelling, a transition flow mode (Re 0-7437) was revealed behind the sliding shutter. As such, it can be assumed that the most suitable place for measuring equipment to be installed is directly behind the closed part of the sliding shutter.

**Keywords:** silicon production; nanoparticles; ore-thermal furnace (OTF); gas cleaning; symmetry; carbon footprint; CFD; ANSYS fluent

## 1. Introduction

A large number of fine particles with various compositions are emitted into the atmosphere during carbothermic silicon reduction. Analyses of waste gases show that most emissions are related to the consumption of carbon materials that contribute to the overall carbon balance. These are carbon-graphite electrodes that are used for heating quartz and charcoal to reduce the amount of silicon released from oxides (quartz) [1–3]. The main component of waste fume emissions is $SiO_2$ microsilica (up to 85 percent), which is present in a mixture of solid carbon in various forms and states (7–8 percent) and in silicon carbide (5 percent) [4,5]. As a rule, industrial emissions are not controlled, and dust is collected from gas pass systems and from deposits on equipment and building structures.

Granulometric analyses of the fumes captured by GCS electro-filters indicate the presence of particles in microsilica fumes that are 200–250 μm in size and have an elevated carbon nanoparticle content (up to 8 percent) that can be removed with the off-gas.

Currently, carbon-free microsilica is widely used as a modifying additive to base materials in the construction industry. Thus, the use of microsilica makes it possible to produce concrete with special properties: increased durability (resistance to the action of weak acids and seawater) and increased compression strength. Production methods for silicon-carbide powder materials, such as micronized carbide (particle size < 1 m) for ceramics and nanocarbides (particle size < 1 nm) for high-quality

structural ceramics and galvanics, are being intensively developed, as they create a high-value of the final product [6,7].

The main sources of exhaust gas carbon emissions are carbon electrodes, activated carbon, and carbonaceous materials.

For electrodes, the main components of carbon flow are arc heating fractures, the main quartz reduction reaction (stoichiometry), the decrease and oxidation of the lateral faces, and the destruction of the electrode soles upon contact with the charge. The key factor in electrode mass erosion is the oxygen concentration in the furnace atmosphere. As a result of the increased formation and oxidation of carbon monoxide, the oxygen concentration in the furnace atmosphere will decrease [8–10].

There is scientific and technical interest in modelling the distribution of waste gas concentration fields during carbothermic silicon reduction. It is necessary to ensure that the furnace's thermal conditions and the balance of consumable carbon are controlled effectively [11].

## 2. Problem Overview

### 2.1. Causes of Carbon Nanoparticle Formation during Carbothermic Silicon Reduction in Ore-Thermal Furnaces

At present, reducing the carbon footprint in pyrometallurgical processes [6] is relevant to the resolution of global decarbonization issues. The manufacturing process in ore-thermal furnaces (OTFs) involves multi-component carbon systems (carbon electrodes and lining, activated charcoal, modifying additives) that affect the overall carbon balance during smelting. Activated carbon is consumed by the main silicon reduction reaction by means of stoichiometry, as well as by the side reactions caused by interactions with the impurities in quartz that take place according to thermodynamic conditions. For example, one of these irrevocable losses can be attributed to the transfer of active carbon into the crust and the subsequent production of silicon carbide when there are temperature disturbances of more than 1700–1900 °C [12]. Additionally, thermal disturbances in the furnace and increased carbon consumption are associated with carbon oxidation (an increase in CO and $CO_2$ content) or with the abrupt release of soot and carbon particles into the atmosphere of the furnace and into the gaseous space [13].

Charge materials are consumed for the main silicon reduction reaction to form silicon dioxide, which may be accompanied by a transition into carbon monoxide in parallel with the formation of intermediate silicon monoxide, as well as silicon carbide during overheating, especially at the start of the smelting process and during the primary arc [14]. A separate consumption item is the unresponsive carbon that is derived from charcoal, which forms nanoparticles in the form of amorphous carbon, active pure carbon, or fullerenes.

The carbon resulting from electrode chipping and falling passes into a fume mixture together with microsilica. In this case, carbon particles without interaction are adsorbed on the highly developed surface of the silica fumes to form microsilica, the main form of silicon waste. This is an item of consumption due to the changes in the silicon balance that range from 380 to 450 kg per ton of produced silicon and requires operational control. In this instance, sampling and relevant data on the chemical composition of the microsilica captured in the GCS are needed for analytical comparison with the source quartz and impurities to ultimately determine the quality of silicon products [15–17].

### 2.2. Microsilica Monitoring and Nanoparticle Capture in the Ore-Thermal Furnace Gas Cleaning System

The main problem with capturing and controlling nanoscale particles in the OTF GSC is the timely determination of the emission composition of the gas fumes (in mass).

Conventional GSC implements a detailed analysis after the end of the electric filter cycle and after the filter has been cleaned and the sediment has been weighed according to the time. As a result, there is a delay in the response when the electric and technological modes of the furnace are disrupted and when the electrodes are burned, which is indicated by an increase in the carbon content in the waste gases [18]. Monitoring the waste composition during the production of silicon and its alloys is associated with a number of industry-specific problems:

- The temperatures of the gases in the immediate vicinity of the furnace roof are very high (600–850 °C), resulting in the need to cool and ventilate the gas-dust flow before it makes contact with the sensitive instruments when taking extractive measurements;
- The high concentration of particulates in front of the filter can cause rapid abrasive damage to the measuring instruments in the gas streams that are in contact with the particles and can also affect the measured data [19,20];
- The turbulent mode of gas flow has a wide range of time and space scales for the pulsations of all of the flow characteristics. This makes the gas flow faster than the laminar flow and results in intensive mass exchanges with high-impulse and energy levels between different stream regions due to the intensive mixing of the dispersed medium. This results in the substance having an uneven distribution in the gas flow and a consequent distortion of the measurement results.

Taking into account the described flow conditions in the furnace atmosphere and in the gas duct system of GCSs, the question of how to determine the temperature and velocity control points of the particles arises. For this purpose, it is necessary to define stable zones on the mass transfer interaction path.

To effectively control the electric mode of the furnace, installing a gas analyzer between the gas duct and the gas purification equipment in the stable flow region of the gas stream is recommended. This allows additional data to be obtained to create a digital database of information about the process [21].

The continuous growth of computing capacities and measuring technologies in recent years has opened up radically new possibilities for the computer modelling of large production units [22].

The novelty of the proposed solution is in justifying the location of the gas analyzer installation with the aid of the created computational fluid dynamics (CFD) model. The ANSYS software package, specifically its CFD modelling module ANSYS Fluent, was used for the mathematical model.

*2.3. Main Features of Flue Gas Movement in the Ore-Thermal Furnace Gas Cleaning System*

In most cases, when taking the fundamental laws of gas-fluid dynamics [4] in the furnace gas duct into account, the flue flow has a high Reynolds value and is turbulent due to different densities.

This movement of gases with dust is accompanied by an intensive mixing with velocity and pressure pulsations, and in addition to the main longitudinal movement, transverse movements and rotational movements are observed in individual flow volumes, especially in the vortex zones that are close to the filters and ejectors. As a result, the local cross-sectional velocities of the entire gas content flow from the furnace vault to the GCS during the mixing process under the influence of a temperature gradient between the layers for the entire mixing period.

It should be noted that turbulence has a continuous influence on the main flow parameters such as the concentration of components, temperature, velocity, and heat state. Thus, each turbulent vortex volume has its own substance and temperature concentration [23].

To avoid components and pulsations mixing and to thus obtain adequate data on flow parameters and the adequacy of the matrix, it is necessary to define data for the

transient flow by taking the measurements into account in practice. On the other hand, the transient flow mode is characterized by the low mixing rate of the pulsation particles compared with the turbulence mode. In this instance, the key value for determining the flow of gas is the Reynolds number (Re) that characterizes the mode changes.

For this process, it has been experimentally determined that the Re for straight, smooth pipes with the most perturbed flow at the entrance of the pipe is 2300. However, at Re values above this value and up to a certain limit, a transient (mixed) flow mode is observed, and after this limit, turbulent flow is more likely [24]. In cylindrical receptacles (pipes), this transition interval can be varied significantly by reducing the initial disturbance of the medium by up to 50,000. For this process, we determined the factors influencing the turbulent state of the system in the gas pass system from the furnace:

- The current state of the fume environment (kinetics, thermodynamics, etc.): compressibility (velocity of motion of silica fume) taking into account multi-phase flow (interphase exchange).
- The outer limits of gas flow by zone: the movement along the side lining into the vault and output into the gas channel without the influence of the channel geometry.
- The process limit stage and GCS outlet (GCS electric filter surface pressure): the flow area on the surface of the electric filter and the pressure.

The Reynolds criteria for the different flow mode models are as follows:

1. At $Re < 2 \times 10^3$

laminar flow is observed.

2. At $Re > 10^4$

the flow becomes turbulent, but when gases begin to exit the furnace vault and move into the gas channel, it is preserved.

3. At $Re > 5 \times 10^4$

a turbulent boundary layer begins to form during the beginning of the gas flow process due to a sharp change in the temperature.

4. At $2 \times 10^4 < Re\ 10^4$

a transient flow and heat exchange mode are observed when the flow approaches the electric filter. Thus, given the turbulent flow conditions, the ratio of the average to maximum velocity increases with the turbulence. In the Re, this ratio is asymptotically close to 1 [25–27].

### 3. Materials and Methods

*3.1. Computer Simulation of the Dispersion Fluid Dynamics*

Offsetting the limit values for transient flows is a major difficulty in determining the flow parameters of a multi-component gas. This problem is particularly pronounced in the case of flow limits since the existing standard models can only estimate the average flow rate of a gas duct when averaging the Reynolds criteria values in ANSYS Fluent [28].

The exhaust ducts and flows of the carbothermic silicon reductions in the OTF were simulated to detect stable flow zones in which the control point for determining the concentration of the exhaust gases could be installed.

The classification of existing methods and approaches for the numerical modelling of turbulent flows is based on the level of detail required to refine and detect turbulent pulsations, as well as their energy spectrum and type of flow. Depending on this, methods can be divided into three groups (DNS—direct numerical simulation, LES—large eddy simulation, and RANS—Reynolds-averaged Navier–Stokes) [29].

For the current task, the best solution to account for large-scale turbulent vortices and flows in the boundary layer is a hybrid approach in which an RANS model is activated in a wall area with an LES model close to it. This is a detached eddy simulation

(DES) method for simulating disconnected vortices that takes into account the reduction in turbulent vortices near a solid surface [30]. This method requires significant computational resources but ensures its adequacy compared to others.

In the considered case, taking the influence of the boundary layer in the gas flow transition zones on turbulence into account, the explicit resolution of even the largest vortices in the boundary layer requires considerable grinding in the calculation grid, resulting in an increase in the calculation time. Thus, a wall-adapted local eddy-viscosity model (WALE) [31] can solve this problem, taking the particularities of the gas flow and its thermal potential into consideration.

The WALE method requires a high-quality grid. The number of grid elements should be more than 1,000,000 at least, and boundary layers should be used for inflation. Otherwise, it is possible that uncorrected data will be obtained as a result of the numerical simulation.

*3.2. Problem Statement: Modelling the Off-Gas Mixture under Ore-Thermal Furnace Conditions*

The symmetry of the gas duct system ensures uniform flow separation. An increase in the waste gas concentration in one part of the duct will result in changes in the pressure drops in another part. This will consequently lead to asymmetric flows, making it impossible to define a stable flow area. Gas duct system symmetry allows us to evaluate the adequacy of the resulting models.

The following boundary conditions (Tables 1–3) were used as input components for modelling the furnace exhaust gas system: the water-cooled roof of the furnace and the steel gas passes. A sliding shutter was placed at a 45-degree angle at the fork of the two gas passes.

**Table 1.** Initial model data.

| Parameter | Unit | Value |
|---|---|---|
| Volume (gas) | $Nm^3/h$* | 1000 |
| Temperature (gas) | °C | 500 |
| Pressure (water) | kPa | 250,000 |
| Volume (water) | $Nm^3/h$* | 250,000 |
| Temperature (water) | °C | 35–45 |

*—$m^3/h$ under normal conditions (0 °C, 100 kPa).

**Table 2.** Average composition of the waste gases in percentages [6].

| Mixture Component | Percent |
|---|---|
| CO | 88.6 |
| $CO_2$ | 4.81 |
| $CH_4$ | 1.42 |
| $N_2$ | 2.5 |
| $H_2$ | 2.67 |

**Table 3.** Average composition of the fumes in percentages [7].

| Mixture Component | Percent |
|---|---|
| $SiO_2$ | 85.41 |
| $Al_2O_3$ | 0.46 |
| $Fe_2O_3$ | 0.30 |
| CaO | 1.50 |
| MgO | 1.24 |
| C | 6.09 |

| | |
|---|---|
| Na$_2$O | 0.08 |
| SO$_3$ | 0.16 |
| P$_2$O5 | 0.12 |
| K$_2$O | 0.31 |
| TiO$_2$ | 0.02 |
| SiC | 5.03 |

The average values of the fumes in the off-gas mixture are 10–12 percent.
According to the goals of this research, two models were developed:

- A system with a water-cooled portion of the roof for gas passes.
- A gas pass system without water cooling.

### 3.3. Governing Equations

The mathematical modelling of turbulence is based on the set of ratios used to describe the movement of a fluid/gas. The basic governing Navier–Stokes equation system consists of the equation of continuity (1) and the equation of motion (2):

$$\frac{\partial \rho}{\partial t} + \frac{\partial \rho \cdot u_j}{x_j} = 0 \tag{1}$$

$$\frac{\partial \rho \cdot u_i}{\partial t} + \frac{\partial \rho \cdot u_j \cdot u_i}{x_i} = -\frac{\partial P}{x_i} + \frac{\partial \tau_{ij}}{x_j}$$

where:

$P$ —pressure of the gas mixture;
$\mu$—gas mixture density;
t—time;
u$_{i,j}$—velocity components in the i and j directions;
$\tau_{i,j}$—shear strain tensor.

To simulate the particulate matter (fume), a discrete phase model (DPM) was developed using the Euler–Lagrange method. It was constructed by solving the time-averaged Navier–Stokes equations for the liquid phase, which was treated as a continuum, while the dispersed phase was described by tracing the trajectory of a large number of particles through the calculated flow fields. The dispersed phase can exchange momentum, mass, and energy with the liquid phase.

The balance of the forces acting on a particle predicts the trajectory of the discrete phase particles by integrating the balance of the forces on the particle, which is written in the Lagrangian reference frame. This force balance equates inertia with the forces acting on the particle and can be written (based on the directions in the Cartesian coordinate system) as:

$$\frac{\partial U_p}{\partial t} = F_D(U - U_p) + \frac{g_x \cdot (\rho_p - \rho)}{\rho_p} + F_x \tag{1}$$

where F$_D$($U - U_p$) is the drag force per unit mass of the particle.

## 4.. Results and Discussion

### 4.1. Model 1: Combination of the Furnace's Roof and Water-Cooled Gas Passes

As initial data, in addition to the variables specified in Tables 1–3 as well as the dimensions, the following parameters were made available: water-cooled thickness, 0.06 m; side lining thickness, 0.05 m; and thickness of the non-water-cooled part of the roof made of chamotte bricks, 0.5 m.

The grid of this model consisted of 2,037,611 elements to obtain adequate simulation accuracy and had a minimum orthogonal quality of at least 0.1 (Figure 1). The LES WALE turbulence model was used to solve the variable-density subsonic flow problem by means of a pressure–based coupled solver. The real Soave–Redlich–Kwong gas model [32,33] was used for the gas mixture.

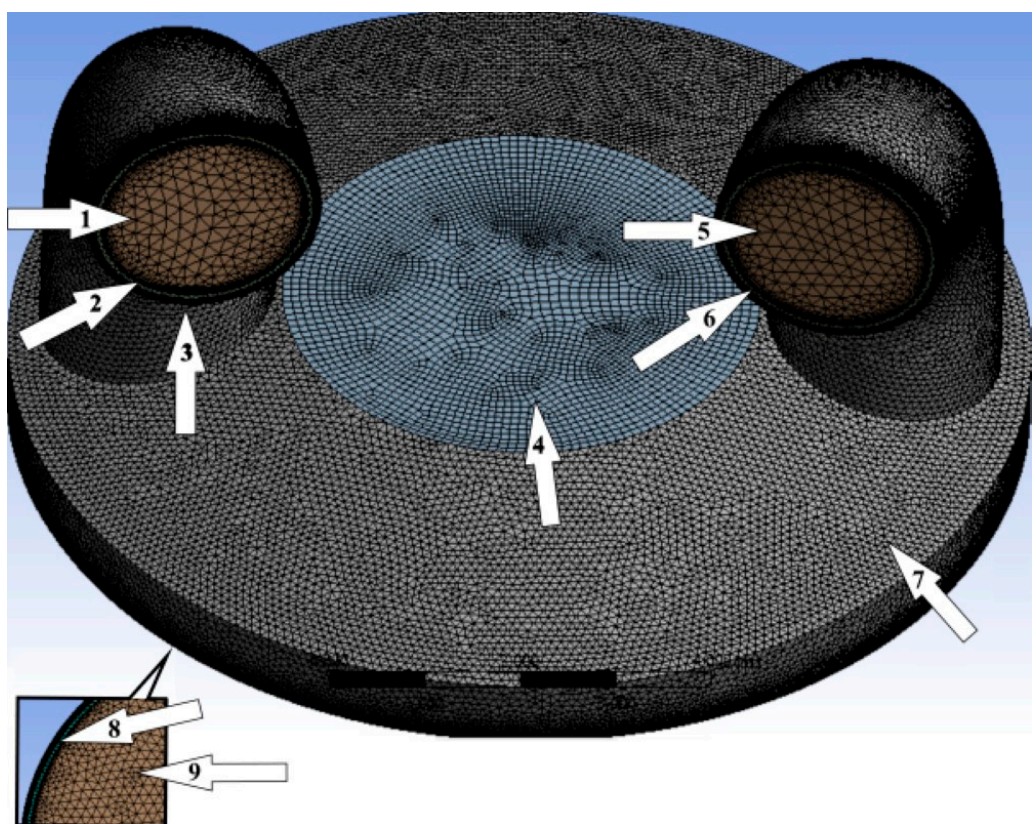

**Figure 1.** Furnace roof model. (1, 5—exhaust gas outputs; 2, 6—cooling water outputs; 3, 7—concrete refractory; 4—chamotte refractory; 8—exhaust gas input; 9—cooling water input.)

The boundary conditions for both streams (exhaust gases and cooling water) are shown in Tables 4 – 6.

**Table 4.** Model 1 boundary conditions for exhaust gas mixture.

| Parameter | Input | Output |
|---|---|---|
| Type of boundary conditions | Mass-flow inlet | Pressure outlet |
| Hydraulic diameter, m | 10.34 | 3 |
| Mass flow rate, kg/s | 29.16 | - |
| Gauge pressure, Pa | - | 0 |
| Temperature, °C | 500 | - |
| Re | 60,889.4 | 236,962.6 |
| Turbulence intensity, percentage | 10.34 | 3 |

**Table 5.** Model 1 boundary conditions for cooling water.

| Parameter | Input | Output |
|---|---|---|
| Type of boundary conditions | Mass-flow inlet | Pressure outlet |
| Hydraulic diameter, m | 0.06 | 0.06 |
| Mass flow rate, kg/s | 278 | - |
| Gauge pressure, Pa | - | 0 |

| Parameter | Input | Output |
|---|---|---|
| Temperature, °C | 35 | - |
| Re | 150,075 | 150,075 |
| Turbulence intensity, percent | 0.06 | 0.06 |

**Table 6.** Model 2 boundary conditions.

| Parameter | Input | Output |
|---|---|---|
| Type of boundary conditions | Mass-flow inlet | Pressure outlet |
| Hydraulic diameter, m | 3 | 2.7 |
| Mass flow rate, kg/s | 15.54 | - |
| Gauge pressure, Pa | - | 0 |
| Temperature, °C | 430 | 0 |
| Re | 117,255.6 | 266,330 |
| Turbulence intensity, percent | 3.72 | 3.35 |

As a consequence of simulation, the temperature distribution and gas velocity profiles were obtained to define the boundary conditions for the second model as the area in which the intended gas analyzer could be installed.

At the outlet from the water-cooled part of the gas pass system, the gas flow rate was 5.5 m/s, which is equivalent to a mass flow rate of 15.54 kg/s. The temperature of the mixture was 450 °C.

Because the model had a horizontal symmetry, we were able to evaluate the temperature and velocity distribution using the cross-sectional area at the centre of one of the water-cooled parts of gas ducts 2 (Figure 2). This model makes it possible to determine the input parameters to assess possible variation among them. In this way, a prediction story was created for the next part of the GCS, in which the steel gas passes through a sliding shutter model. This provides stable initial characteristics for further modelling tasks.

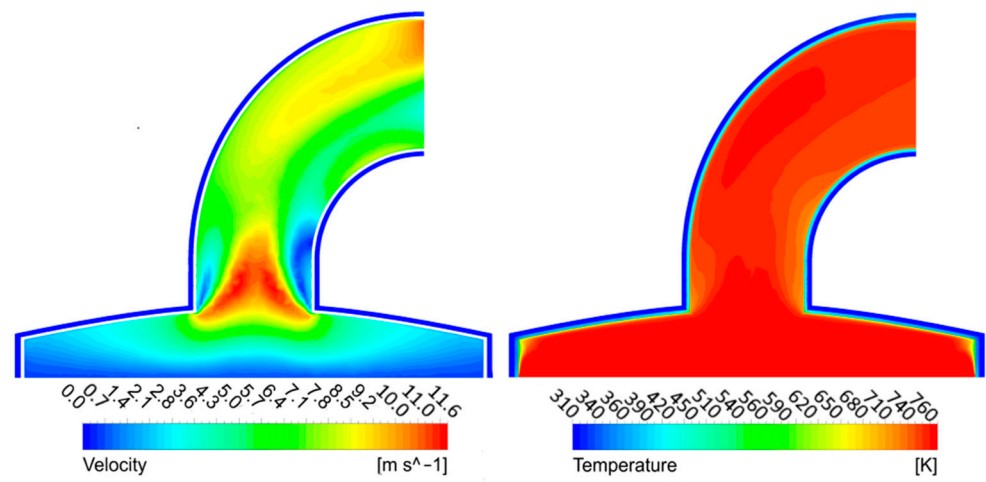

**Figure 2.** Sectional view of temperature and velocity distribution contours of model 1.

*4.2. Model: 2 Gas Passes without Water-Cooling*

The simulated part of the exhaust gas system consists of gas passes with a rigid steel frame and a sliding shutter, which acts as a gas velocity regulator and is located at the gas pass junction. This part is of greatest interest for modelling and further analysis because stable zones were predicted to be here.

There were 2,739,629 grid elements in this model (Figure 3). The grid had an acceptable average orthogonal quality of 0.79.

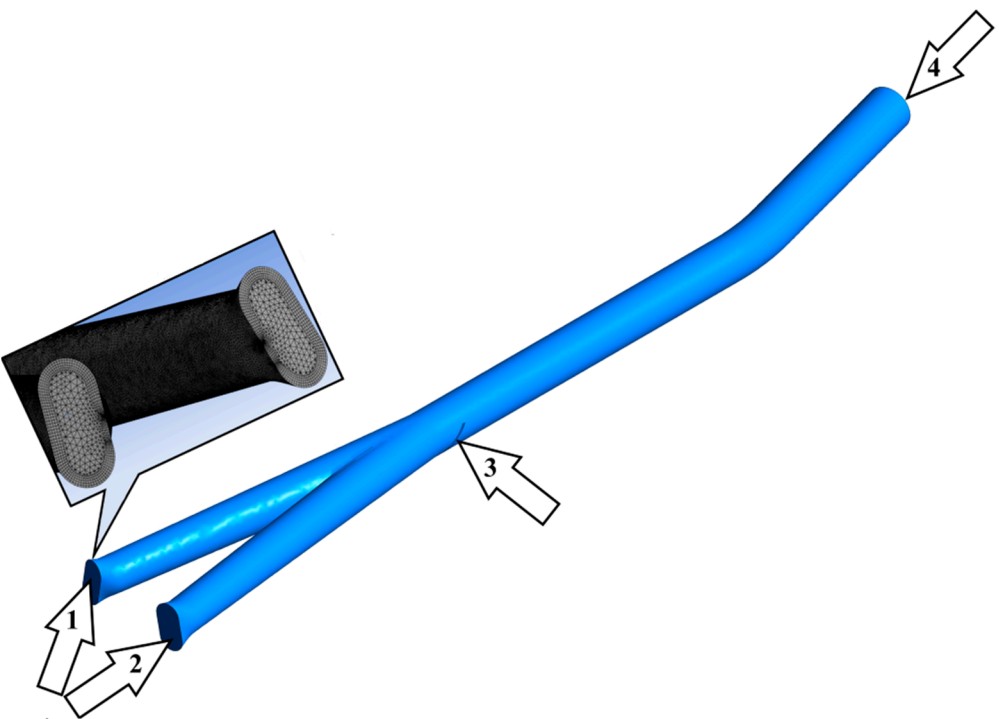

**Figure 3.** Pass model. (1, 2—exhaust gas inputs; 3—sliding shutter; 4—exhaust gas output.)

Based on the fact that the most stable flows in the models will not be laminar ones because of the high velocities, considering the transient mode that was identified by means of ANSYS Fluent is advised. The Re ratio was calculated using the conventional equation:

$$\text{Re} = \frac{u \cdot d_{mix} \cdot \rho_{mix}}{\mu_{mix}} \tag{2}$$

where:
u—velocity, m/s;
$d_{mix}$—hydraulic diameter, m;
$\rho_{mix}$—mixture density, kg/m$^3$;
$\mu_{mix}$—dynamic viscosity of the gas mixture, *Pa s*;
$d_{mix}$—hydraulic diameter, m;
$\rho_{mix}$—mixture density, kg/m$^3$;
$\mu_{mix}$—dynamic viscosity of the gas mixture, *Pa s*.
Sutherland's formula was used to determine the dynamic viscosity:

$$\mu_t = \mu_0 \cdot \frac{273 + C}{T + C} \cdot \frac{T^{\frac{3}{2}}}{273} \tag{3}$$

where $\mu_t$—dynamic viscosity of the gas at the temperature, *Pa s*;.

$\mu_0$—dynamic viscosity of the gas at 0 °C, *Pa s*;

T—the absolute temperature of the gas, K;

C—Sutherland's constant.

Thus, the dynamic viscosity of the gas mixture can be found using the following equation:

$$\frac{M_{mix}}{\mu_{mix}} = \frac{a_1 \cdot M_1}{\mu_1} + \frac{a_2 \cdot M_2}{\mu_2} + \dots + \frac{a_n \cdot M_n}{\mu_n} \tag{4}$$

where $M_{mix}$, $M_1$, $M_2$, and $M_n$ are the molecular masses of the gas mixtures and their components; $a_1$, $a_2$, and $a_n$ are the volumetric fractions of the components in the gas mixtures; and $\mu_{mix}$, $\mu_1$, $\mu_2$, and $\mu_n$ are the dynamic viscosities of the gas mixtures and their components, $Pa \cdot s$ [34,35].

The density of the furnace gas mixture is a very important indicator for determining the concentration field profile that is repeated on the GCS electric filters:

$$\rho_{mix} = y_1 \cdot \rho_1 + y_2 \cdot \rho_2 + \dots + y_n \cdot \rho_n \tag{5}$$

where $y_1$, $y_2$, and $y_n$ are the volumetric fractions, and $\rho_1$, $\rho_2$, and $\rho_n$ are the densities of the components, kg/m$^3$.

$$\rho_0 \cdot \frac{273}{273 + t} \tag{6}$$

The kinematic viscosity was calculated using the following formula:

$$\upsilon = \frac{\mu_{mix}}{\rho_{mix}} \tag{7}$$

As a result of calculating the required auxiliary parameters in ANSYS CFD-Post, the contours of the main off-gas parameters, such as velocity and kinematic viscosity, were obtained. These parameters have a direct impact on the Re as a key criterion for the flow conditions.

Any obstacle in the gas flow path changes the dynamic flow characteristics. As such, it is necessary to know what happens before and after the gas flow encounters the sliding shutter to evaluate any changes in the cross-section of the gas path. Knowing this, we can determine the control actions for the system.

Figure 4 illustrates the variations in the dynamic characteristics of the gas-dust flowover and the volume of the gas duct. A transient flow mode can be observed in the volume-rendering region behind the closed part of the sliding shutter (shown by the pointer) with the smallest Re. Taking into account the initial conditions of developing the best possible environment (laminar flow) for measurements to be taken in because of the high velocities, the transition mode revealed here can be considered to be an appropriate result of the modelling.

The flow velocity is the most significant parameter affecting both the flow mode and Re. The linear dependence of the Re on the flow velocity, as shown in Figure 5, allows us to estimate the flow mode using the volumetric velocity profile alone for future calculations. Based on this chart, the appropriate velocity should be less than 1 m/s to reach the transient Re.

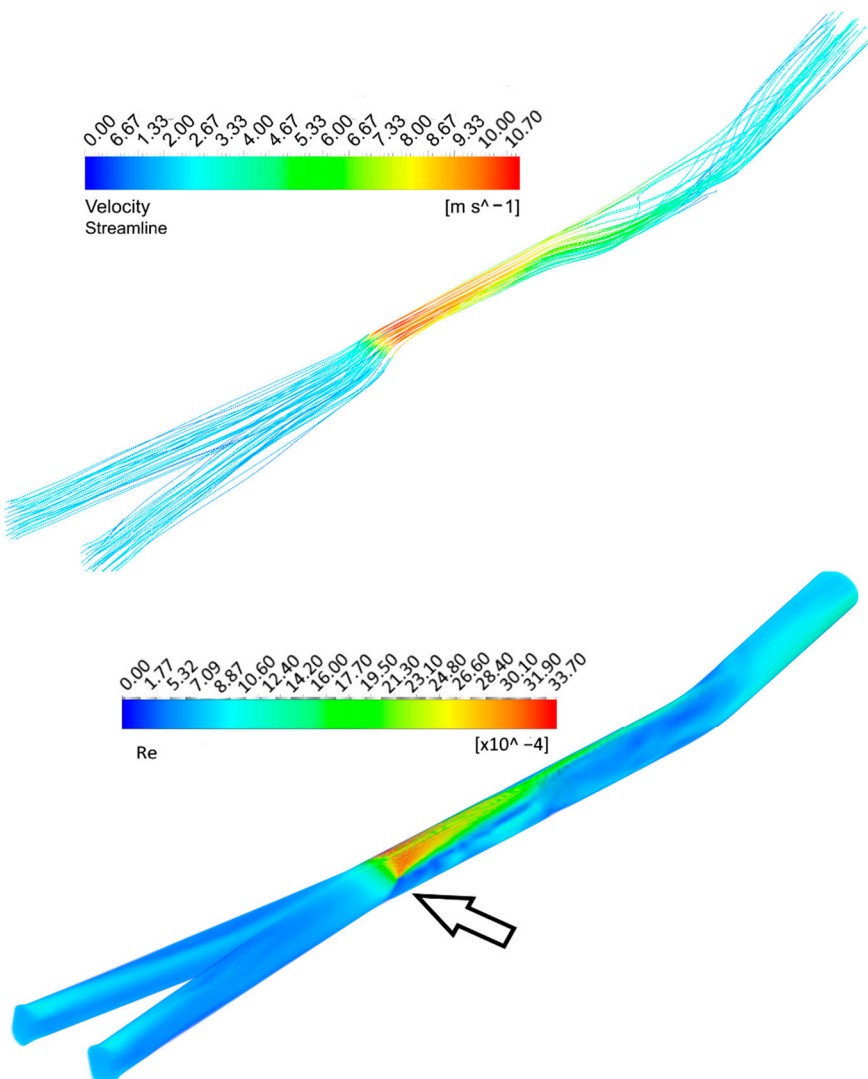

**Figure 4.** Velocity and Re distribution contours in the gas passes.

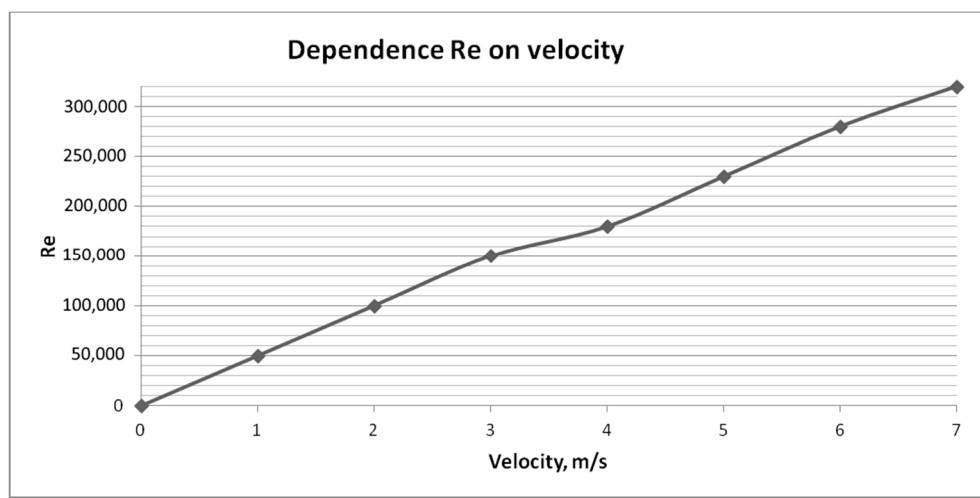

**Figure 5.** Re/velocity dependence chart.

Histograms, such as those in Figure 6, are needed to confirm the existence of a transitional mode and to estimate the proportion of its volumetric distribution in particular figures. A small fraction of transient off-gas flow (red column on the histogram (0-7437) Re) can be observed. These values are appropriate for minimizing turbulent pulsations in this case.

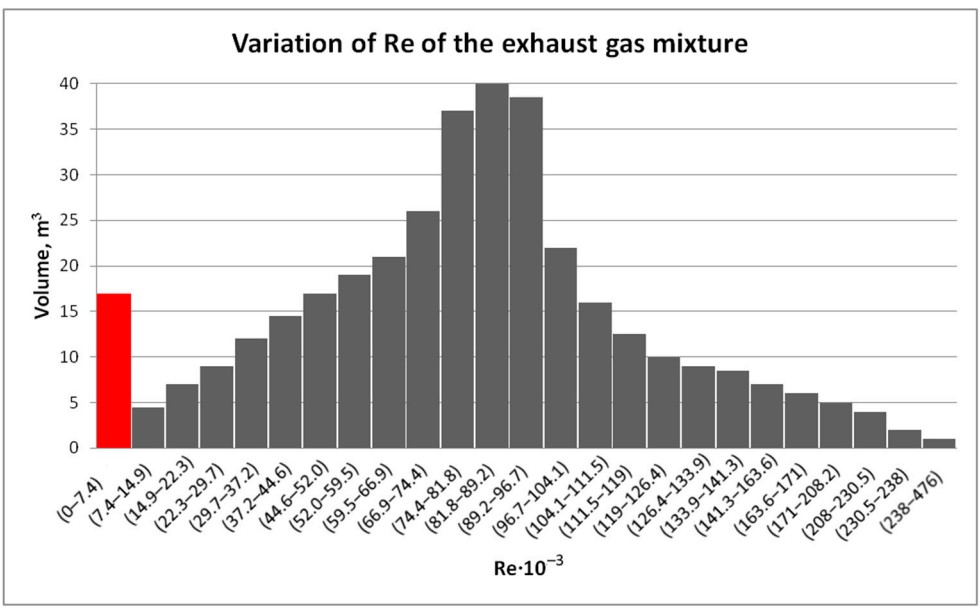

**Figure 6.** Histograms showing the variations in the Re of the exhaust gas mixture.

Thus, despite the wide variety of turbulence models in ANSYS Fluent, in cases where there is strong turbulence, considering transient flow modes is advised.

## 5. Conclusions

The silicon smelting process in OTFs is only controlled in the furnace and at the inlets and outlets of the gas duct. Thus, everything that goes into the gas ducts is a "black box". Emissions can only be controlled when the electrostatic precipitators are clogged. Modelling allows us to look inside the process and to react to the changes that take place in a short period of time.

Something else that is important for process control is minimizing the effects of turbulent pulsations on measurements in the gas duct. The main innovation of this paper is the determination of stable zones by means of CFD modelling to avoid measurement errors. All of the methods mentioned above can profoundly increase the transparency and controllability of the silicon production process.

When producing silicon in an OTF when the exhaust gas was at a volume flow rate of 250,000 m³/h (under normal conditions), strong turbulence developed in the gas ducts, preventing the adequate measurement of the concentrations of each gas mixture component. In order to control the velocity at the fork of the two gas passes, a sliding shutter was installed in one of them at a 45° angle. This resulted in a 4-5-fold reduction in the velocity at the exit of the gas duct. A transient flow mode (Re < 10,000) was formed directly behind the closed part of the shutter, allowing the concentrations of the flue gas components to be measured with the required accuracy and to be controlled using an additional parameter.

In modern conditions, the production of silicon from quartz raw materials in OTFs needs to address energy efficiency issues by taking into account the distribution of gas-fume streams. As a result of the simulation presented here:

- The contours of the main parameters defining the flow mode in the exhaust gas transfer line, namely the kinematic viscosity and velocity, were obtained.

- The flow mode was determined by calculating the Reynolds criterion along the exhaust gas transfer line from the OTF to the GCS.
- It was revealed that the most suitable place for the installation of measuring equipment is directly behind the closed part of the sliding shutter. In this area, there is a transient flow mode with the lowest velocity and lowest Reynolds criterion value. In this location, the flow is influenced by turbulent forces at least, allowing the concentrations of the flow components to be measured with the required accuracy.

**Author Contributions:** The conceptualization and methodology of this research have been developed by V.B.; the model development and validation have been performed by O.M. As for writing, the original text has been written by O.M.; editing of the article, as well as supervision, has been done by V.B. All authors have read and agreed to the published version of the manuscript.

**Funding:** This research was funded by Russian Science Foundation grant no. 22-29-00397.

**Institutional Review Board Statement:** The study was conducted according to the guidelines of Saint Petersburg Mining University and approved by the University Review Board (or Ethics Committee) of Saint Petersburg Mining University.

**Data Availability Statement:** The results was obtained during the authours investigation and were not published in other papers.

**Conflicts of Interest:** The authors declare no conflict of interest.

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
