# Peer review of "Monitoring of the Behaviour and State of Nanoscale Particles in a Gas Cleaning System of an Ore-Thermal Furnace"

_symmetry, doi:10.3390/sym14050923_

Round 1
Reviewer 1 Report
The current work investigated behavior and state of nanoscale particles in a gas cleaning system of an ore-thermal furnace. Generally, the topic seems interesting. However, some fetal problems exist. I do not recommend its publication in the current form.
1. The whole manuscript were not well written, e.g., the abstract is suggested to be re-written and the last sentence of the abstract could be canceled (The study is related to the direction of “Energy Efficient Technologies” of Saint Petersburg Mining University and carried out within the framework of the Russian Science Foundation Grant no. 22-29-00397).
2. Since the main work was done by numerical simulation, and some key issues towards numerical simulation should be presented, e.g., grid independence.
3. The quality of the figures should be improved, e.g., Fig.5 and Fig.6.
Author Response
Thank you for your review! We tried to improve our paper in according to your advices.
Please see an attachment.

Reviewer 2 Report
Queries and suggestions are attached.

Author Response
Thank you for your review! We tried to improve our paper in according to your advices.
Please see an attachement.

Round 2
Reviewer 1 Report
There is some improvement comparing with last version. However, some problems should still be addressed.
1. The abstract should be even more concise to present main works of the current manuscript.
2. The results done by simulation should be discussed and analyzed.
3. What is the main innovation of the current work, refering with existing works?
Author Response
Thank you for a great contribute to our researc! We have revised our paper according to your comments as well as improve English language having used MDPI ediditing service
Please see the attachment.
